# Impact of Sarcopenia and Frailty in a Multicenter Cohort of Polypathological Patients

**DOI:** 10.3390/jcm8040535

**Published:** 2019-04-18

**Authors:** Máximo Bernabeu-Wittel, Álvaro González-Molina, Rocío Fernández-Ojeda, Jesús Díez-Manglano, Fernando Salgado, María Soto-Martín, Marta Muniesa, Manuel Ollero-Baturone, Juan Gómez-Salgado

**Affiliations:** 1Internal Medicine Departments, Hospital Universitario Virgen del Rocío, 41013 Sevilla, Spain; aglezmolina@gmail.com (Á.G.-M.); m.ollero@telefonica.net (M.O.-B.); 2Hospital San Juan de Dios del Aljarafe, 41930 Sevilla, Spain; MariaRocio.Fernandez@sjd.es; 3Hospital Royo Villanova, 50015 Zaragoza, Spain; jdiez@aragon.es; 4Hospital Regional, 29010 Málaga, Spain; fersalord@gmail.com; 5Hospital Juan Ramón Jiménez, 21005 Huelva, Spain; msoto@hotmail.com; 6Hospital San Juan de Dios de Pamplona, 31006 Pamplona, Spain; mmuniesa@ohsjd.es; 7Department of Sociology, Social Work and Public Health, Universidad de Huelva, 21071 Huelva, Spain; jgsalgad@gmail.com; 8Safety and Health Posgrade Program, Universidad Espíritu Santo, Guayaquil 091650, Ecuador

**Keywords:** polypathological patients, multimorbidity, sarcopenia, frailty

## Abstract

The prevalence, relationships and outcomes of sarcopenia and frailty in polypathological patients remain unknown. We performed a multicenter prospective observational study in six hospitals in order to assess prevalence, clinical features, outcome and associated risk factors of sarcopenia and frailty in a hospital-based population of polypathological patients. The cohort was recruited by performing prevalence surveys every 14 days during the inclusion period (March 2012–June 2016). Sarcopenia was assessed by means of EWGSOP criteria and frailty by means of Fried’s criteria. Skeletal muscle mass was measured by tetrapolar bioimpedanciometry. All patients were followed for 12 months. Factors associated with sarcopenia, frailty and mortality were analyzed by multivariate logistic regression, and Kaplan–Meier curves. A total of 444 patients (77.3 ± 8.4 years, 55% males) were included. Sarcopenia was present in 97 patients (21.8%), this being moderate in 54 (12.2%), and severe in 43 (9.6%); frailty was present in 278 patients (62.6%), and 140 (31.6%) were pre-frail; combined sarcopenia and frailty were present in the same patient in 80 (18%) patients. Factors independently associated to the presence of both, sarcopenia and frailty were female gender, older age, different chronic conditions, poor functional status, low body mass index, asthenia and depressive disorders, and low leucocytes and lymphocytes count. Mortality in the 12-months follow-up period was 40%. Patients with sarcopenia, frailty or both survived significantly less than those without these conditions. Sarcopenia and frailty are frequent and interrelated conditions in polypathological patients, shadowing their survival. Their early recognition and management could improve health-related outcomes in this population.

## 1. Introduction

The improvement in life conditions, as well as in the social and health systems, has led to an increase in life expectancy in most of the world. As a matter of fact, in Europe, the percentage represented by citizens over 65 years will rise from 18.9% in 2015 to around 29% in 2080 [1]. We live longer, and survive to many diseases, but the cost we are paying is the increase of chronic conditions, which accompany us in the last years of life.

This new social and epidemiological trend is becoming of nuclear importance in establishing medium- and long-term strategies in most healthcare systems. A rational and appropriate management of multimorbidity is considered nowadays critical to maintain our systems viable [2]. In the area of chronic conditions, polypathological patients (PP) are nowadays the clinical paradigm of the emergence of multimorbidity in our societies. They meet all criteria of the so-called ’populations with complex chronic diseases’: they are prevalent in most clinical arenas, they are advanced-aged (mean age of PP in multicenter cohorts rounded 75–80 years), complex, clinically vulnerable, they are prone towards functional decline, and suffer high mortality rates [3,4,5].

Aging is associated with sarcopenia, a geriatric syndrome closely linked to the development of physical disability, loss of quality of life and death. It is defined by a progressive and widespread loss of skeletal muscle mass and strength [6]. Prevalence of sarcopenia increases with age: after reaching 50 years of age, muscle mass decreases 1 to 2% per year, and muscle strength has an annual decrease of 1.5% between 50 and 60 years age, and then 3% each year. Between 5% and 13% of people aged between 60 and 70 years of age and 11% to 50% of people aged 80 or older have sarcopenia [7]. The prevalence of sarcopenia is higher when chronic diseases and/or different organ failures are present; however, its impact and main features in hospital-based PP populations remain unknown.

Another syndrome closely related to aging and sarcopenia is frailty. Frailty is characterized by age-associated decline in physiologic reserve and function across multi-organ systems, leading to increased vulnerability and adverse health-related outcomes [8,9,10]. Although a clear definition remains still elusive, many consider frailty as a wasting syndrome of old age that leaves a person vulnerable to falls, functional decline, morbidity and mortality. Alternatively, it is also defined as a geriatric syndrome of increased vulnerability to environmental factors with underlying physiological mechanisms related to chronic inflammation, many diseases, hormonal adjustments, sarcopenia and vitamin deficiencies [10,11]. Its prevalence is estimated in 7%–8% of community dwelling old adults [12,13]. In the same way as with sarcopenia, the prevalence and main features of frailty among in-hospital PP remain unknown.

For all these reasons, we have developed a multicenter cohort study in order to assess the prevalence, the main features, and the factors associated with sarcopenia, frailty and 12-month mortality in hospital-based PP.

## 2. Patients and Methods

This was a prospective observational, multi-institutional study carried out by researchers from the Polypathological Patient and Advanced Age Study Group of the Spanish Society of Internal Medicine. The study was approved by the ethics committee of all participant centers. The study inclusion period ranged from January 2012 to March 2016.

### 2.1. Reference Population

All patients hospitalized or attended as outpatients in the Internal Medicine and Geriatric areas (in-hospital, as well as in outpatient clinics) from the 6 Spanish hospitals (2 tertiary teaching centers and 4 secondary/basic general hospitals) participating in the study (all participant centers are listed on the PROTEO Researchers list). 

### 2.2. Sample Conformation

The patients’ samples were collected by performing prevalence surveys in the hospital wards, and in the outpatients clinics every 14 days during the recruitment period. Globally, the sample universe summing all participant hospitals rounded 300–400 possible candidates in every prevalence survey. All patients who met inclusion criteria and without exclusion criteria were asked to enter the study. A total of 155 surveys were performed (29 ± 19 surveys per hospital).

### 2.3. Inclusion Criteria

Patients ≥18 years old, who met criteria of PP were included, after providing their written informed consent. A detailed description of the criteria of PP is detailed in Table 1. Patients with implanted metal devices (pacemakers, osteoarticular prostheses…) or with any extremity amputation (because interference with electric bioimpedanciometry methods), those in clinical agony, and those who did not agree to participate in the study, were excluded.

### 2.4. Development of the Study, Data Collection and Follow-Up

After receiving informed consent, a complete set of demographical, socio-familial, clinical, functional, laboratory, and pharmacological data were collected from all included patients.

Demographic and socio-familial data included age, gender, residence, employment data¸ the need for a caregiver, and the main caregiver´s profile. Clinical data included the different diseases, and all possible comorbidities, stage of different diseases (NYHA class and mMRC dyspnea score [14,15], and Child–Pugh stage [16]), assessment of Charlson´s comorbidity index [17,18], different symptoms and signs, body mass index (BMI), functional parameters (Barthel’s index [BI]), and number of hospital admissions in the last 12 and three months, respectively.

Laboratory data included plasma creatinine (Cr (mg(dL)), sodium (Na (mEq/L)), bilirubin (Bb (mg/dL), albumin (ALB (g/dL), hemoglobin (HB (g/dL)), leucocytes (number/µL), lymphocytes (number/µL), cholesterol (mg/dL), triglycerides (mg/dL), vitamin D (ng/mL), cholinesterase (IU/mL), and ferritin (ng/mL). Pharmacological data included the number of chronically prescribed drugs at baseline.

Hand grip strength was assessed by hand-held dynamometry (HHD) (Dynamometer Kern MAP-BA-s-0910, Balingen, Germany) and skeletal muscle mass by tetrapolar bioimpedanciometry (BIA) (BIA 101 AKERN, Pontassieve, Italy; with resistance measure range of 0–999 Ohms, and reactance measure range of 0–100 Ohms). Both were performed to all patients in best clinical stability conditions (at discharge for in-hospital patients, or in the office for stable outpatients); additionally, those patients who were able to walk were asked to perform the 4-meter gait speed test. For HHD, we followed Roberts et al. recommendations with slight modifications (the highest value of four attempts (two per hand), with maximum isometric effort, was noted) [19]. BIA was performed following manufacturer’s guidelines (patients were asked to be in horizontal position for at least 2 minutes to allow an homogeneous distribution of body fluids; bioelectrical values of the tissues were obtained between the bony prominences of the wrist and ankle (metatarsal-metacarpal area) of the right hemisoma, positioning electrodes in hand and foot with a minimal distance of 5 cm between them). Then, resistance and reactance in Ohms were obtained; after this, the standard calculations detailed by the Manufacturer’s instruction and software were performed in order to obtain the skeletal muscle mass [20,21]. Gait speed assessment was performed following the European Working Group on Sarcopenia in Older People EWGSOP recommendations, specifically those used by Laurentani et al. [22]. We only modified this protocol substituting the photocells with a 4 m colored adhesive ribbon, which was adhered to the floor. Participants were instructed to stand with both feet touching the ribbon start and to begin walking at their usual pace after a verbal command, until the end of the ribbon. The time between the start and end was recorded by an investigator with a digital chronometer. The average of two walks was used to compute a measure of walking speed.

All these data were collected by the clinicians in charge of the patients, who were active members of the investigation team. All patients were followed during a 12-month period in order to assess mortality and health-care needs (hospital admissions and number of in-hospital days). Time survival was assessed and, in the case of death, chronology of the demise was incorporated. Therefore, we looked at mortality as both a dichotomous and a time-dependent end-point. For the dichotomous outcome, subjects were categorized depending on whether or not they survived 12 months from their initial interview date. For the continuous outcome, survival time was defined as the number of days between the baseline interview and the date of death.

### 2.5. Sarcopenia Assessment

Sarcopenia was defined following EWGSOP criteria [6]. This was established by the presence of a gait speed ≤0.8 m/s, plus a skeletal muscle mass <6.76 kg/m^2^ in women, and <10.76 kg/m^2^ in men (for those patients able to walk); or a hand grip strength lower than the 50th percentile of is/her age group and gender (in most cases <30 kg in men and <20 kg in women as detailed in EWGSOP recommendations), and the same criteria with respect to skeletal muscle mass (for those patients unable to walk). Moderate sarcopenia was defined in the presence of slow gait speed/grip strength, and a skeletal muscle mass of 5.76–6.75 kg/m^2^ in women, and 8.6–10.75 kg/m^2^ in men; and severe sarcopenia in the presence of slow gait speed/grip strength, and an skeletal muscle mass ≤5.75 kg/m^2^ in women, and ≤8.5 kg/m^2^ in men.

### 2.6. Frailty Assessment

Frailty was defined following Fried’s criteria (slowness, weakness, weight loss, exhaustion, and low physical activity) [8]. We measured every item as follows: slowness was defined using the 4-m walking speed test, considering slowness if the patients performed it in >5 s; weakness was measured by grip strength in the dominant hand using the dynamometer Kern MAP-BA-s-0910, considering weakness if strength values were lower than the 50th percentile of is/her age group and gender; weight loss was considered positive for reporting more than 4.5 kg of unintentional weight loss in the previous year; exhaustion was assessed using two questions (“I felt that anything I did was a big effort” and “I felt that I could not keep on doing things”), considering exhaustion when one or both answers were positive; and low physical activity was based on the Physical Activity Scale for the Elderly [23], those in the worse quintile of physical activity were considered positive for this item. Subjects were classified as frail if they met three or more of Fried’s Criteria [8], as pre-frail if subjects met one or two criteria, and non-frail or robust if none item was present [8,24].

### 2.7. Additional Definitions

Obesity was defined as BMI ≥30 following World Health Organization cut-offs; hypoalbuminemia was defined as albumin levels <3.5 g/dL (severe when <1.8 g/dL, moderate when 1.8–2.69 g/dL, and slight when 2.7–3.5 g/dL); Polypharmacy was defined as the chronic prescription of ≥5 drugs. Dependence on functional status for ADL was defined by a BI <60 points. The need for a caregiver was defined when the patient was functionally dependent (BI < 60) and/or cognitively impaired (Pfeiffer Questionnaire ≥3 errors).

### 2.8. Statistical Analysis

The dichotomous variables were described as whole numbers and percentages, and the continuous variables as mean and standard deviation (or median and interquartile rank in those with no criteria of normal distribution). The distribution of all variables was analyzed with the Kolmorogov–Smirnov test. Possible differences in the presence of sarcopenia and frailty were firstly investigated performing the chi-square test (with the Yates correction and, when necessary, the Fisher exact test), the Student’s *t* for normally distributed quantitative variables, and the Mann–Whitney *U* test in the case of quantitative variables that were not normally distributed. Then, we performed a multivariate backward logistic regression analysis with those variables with differences in the bivariate analysis, in order to know those independently associated to sarcopenia and frailty. With respect to mortality, we included the factors that showed statistical differences in bivariate analysis, in a Cox regression model in order to obtain those independently associated with death. The strength of associations was quantified by calculating *odds ratio* using 95% confidence intervals.

Finally, we also performed Kaplan–Meier curves (and log-rank test), considering death as a time-dependent variable, to assess differences in survival trajectories according to the presence of sarcopenia, frailty or both. Statistics were performed using the SPSS 22.0 software (22.0 Version, IBM, Armonk, NY, USA).

### 2.9. Ethics Committee Approval

The present study has been approved by the ethics committee of all participant centers.

## 3. Results

We included 444 patients with a mean age of 77.3 ± 8.4 years. Fifty-five percent were male. Most of the patients (84.5%, *n* = 375) were included at hospital discharge. Most of them were living at home, and only 6.3% were living in nursing homes. The main clinical features and biological parameters of the recruited patients are detailed in Table 2 and Table 3.

Sarcopenia was present in 97 patients (21.8%). This was moderate in 54 (12.2%), and severe in 43 (9.6%) patients. Sarcopenia was diagnosed by the presence of a slow gait speed plus a low skeletal muscle mass in 23 (5.2%) patients, and by low hand grip strength plus a low skeleletal muscle mass in 74 (16.6%) patients, respectively. Factors independently associated with the presence of sarcopenia were older (*p* < 0.016; odds ratio (OR) 1.05 [1.009–1.089] per every increase of one year of age), masculine gender (*p* < 0.0001; OR 6 [3–12]), chronic lung diseases (*p* = 0.003; OR 2.6 [1.3–4.9]), chronic neurological diseases (*p* < 0.0001; OR 3.9 [2–7]), neoplasia (*p* = 0.03; OR 2.3 [1.1–4]), lower BMI (*p* < 0.0001; OR 1.2 [1.1–1.26] per every decrease of 1 kg/m^2^), astenia (*p* = 0.025; OR 2.1 [1.1–4]), number of hospital admissions in the last three months (*p* < 0.006; OR 1.6 [1.1–2.2] per every hospital admission), and a lower basal BI (*p* = 0.001; OR 1.2 [1.07–1.3] per every 10 points of decrease).

Frailty was present in 278 patients (62.6%), and 140 (31.6%) were pre-frail. The features of frailty and its dimensions are detailed in Figure 1. Factors independently associated with the presence of frailty were female gender (*p* < 0.0001; OR 2.9 [1.7–4.8]), chronic neurological diseases (*p* < 0.014; OR 2 [1.15–3.4]), a higher number of polypathology categories (*p* = 0.005; OR 1.7 [1.2–2.3] per every increase of 1 category), asthenia (*p* < 0.0001; OR 3.46 [2–5.8]), pressure ulcers (*p* < 0.014; OR 2 [1.15–3.4]), chronic pain (*p* < 0.005; OR 2 [1.23–3.2]), anxiety disorders (*p* = 0.012; OR 3.3 [1.3–8.3]), and a lower basal BI (*p* < 0.0001; OR 1.4 [1.3–1.5] per every 10 points of decrease).

Combined sarcopenia and frailty was present in the same patient in 80 (18%) patients. Factors independently associated with the presence of both were female gender (*p* < 0.0001; OR 5.9 [2.6–13]), older age (*p* = 0.028; OR 1.05 [1.005–1.1] per every year), chronic lung diseases (*p* = 0.041; OR 2.1 [1.03–4.5]), chronic neurological diseases (*p* = 0.007; OR 2.8 [1.32–5.8]), neoplasia (*p* = 0.038; OR 2.4 [1.04–5.5]), a higher number of comorbidities different than polypathology criteria (*p* = 0.013; OR 1.23 [1.05–1.5] per every increase of 1 comorbidity), a lower BMI (*p* < 0.0001; OR 1.2 [1.1–1.3] per every point decrease), asthenia (*p* = 0.004; OR 3.3 [1.5–7.4]), depressive disorders (*p* = 0.013; OR 6.9 [1.5-31]), a lower leucocyte count (*p* = 0.019; OR 1.06 [1.01–1.1] per every decrease of 1000 leucocytes), a lower lymphocyte count (*p* = 0.026; OR 1.032 [1.004–1.06] per every decrease of 100 lymphocytes), and a lower basal BI (*p* < 0.0001; OR 1.2 [1.1–1.35] per every 10 points of decrease).

Mortality in the 12-months follow-up period was 40% (*N* = 178). Factors independently associated with mortality were cerebrovascular disease (*p* = 0.033; Hazard Ratio [HR] 1.6 [1.04–2.4], a higher PROFUND index (*p* = 0.006; HR 1.06 [1.02–1.1] per every increase of one point in the index), number of hospital admissions in last three months (*p* = 0.011; HR 1.2 [1.05–1.5] per every hospital admission), a lower lymphocyte count (*p* = 0.006; HR 1.09 [1.03–1.2] per every decrease of 100 lymphocytes/µL), and a lower Barthel index (*p* < 0.0001; HR 1.2 [1.04–1.3] per every decrease of 10 points). The cumulative time-dependent survival rate in patients with sarcopenia, frailty, both, or none of both, along the whole follow-up period, is detailed in Figure 2. Robust patients (without sarcopenia and frailty) survived a mean time of 316 ± 8 days; patients with sarcopenia 288 ± 27 days; those with frailty 252 ± 10 days, and finally those with frailty and sarcopenia survived 240 ± 18 days (*p* < 0.0001).

Hospital admissions and number of in-hospital days in the follow-up period were 1.6 ± 1.5 and 12 ± 13 days per patient, respectively.

## 4. Discussion

In the present study, we have observed a high prevalence of sarcopenia (more than one fifth), and a notably high prevalence of frailty (more than 90% of patients were frail or pre-frail) in hospital-based PP. These results are in concordance with other vulnerability parameters already detected in this population like multiple medical care needs, polypharmacy, poor functional and nutritional status, and propensity to delirium, which are also prevalent in these patients [3,4,5,25,26]. Hence, these data confirm that PP are a paradigm of complex geriatric patients. In the same way, and considering the demonstrated benefits of preventing and treating sarcopenia and frailty in the elderly, these interventions will probably have similar benefits in PP, as those observed in other populations [27,28,29,30].

The prevalence of sarcopenia in PP in the present study was higher to that detected recently in other elderly populations [31,32], which is around 10%. Some authors have questioned the EWGSOP criteria used in our study because they are less strict as the Asian Working Group for Sarcopenia criteria, and the International Working Group for Sarcopenia criteria, and may include more patients in earlier stages of muscle weakness-wasting [31,32]. This could explain the prevalence differences observed in our study; nevertheless, we think that the main reasons that explain the higher prevalence of sarcopenia in our patients are their advanced and complex chronic conditions. As a matter of fact, the disease burden of our patients was as high as a mean of eight chronic diseases per patient (summing up polypathology criteria plus other comorbidities).

With respect to the actual controversy in selecting the optimal definition criteria for sarcopenia, recently, the multicenter FNIH cohort study empirically proposed new cut-off points of weakness and low skeletal muscle mass adjusted by body mass index [33,34]; these new criteria have resulted in a more strict definition, selecting patients with deeper weakness and lower muscle mass. The question about which criteria should be used probably rely on the objectives after identifying sarcopenic patients. If the main objectives were establishing population based interventions (exercise, dietary interventions, nutritional supplementation…) and measures to avoid sarcopenia progression, a more sensitive set of criteria (like EWGSOP criteria) would be more suitable. On the other hand, if the main objectives were performing clinical trials in order to test the efficacy and safety of new expensive drugs, a more specific set of criteria (like FNIH criteria) would be probably more appropriate [35]. For these reasons, many clinicians still use the EWGSOP criteria [36,37,38]. Very recently, the EWGSOP updated the original definition and sarcopenia criteria (EWGSOP2). These new set of criteria emphasize on low muscle strength (hand grip strength 27< kg in men and 16< kg in women) as a key characteristic of sarcopenia, using low muscle quantity (a skeletal muscle mass <7 kg/m^2^ in men and <6 kg/m^2^ in women) and quality to confirm the diagnosis. These cut-off points are significantly stricter than those of EWGSOP 2010 criteria [6]. They also focus on poor physical performance as indicative of its severity. In addition, more clear and uniform cut-off points for the different measures are stated [39]. In this sense, when applying these new EWGSOP2 criteria set to our cohort, the prevalence of sarcopenia would have decreased to 2.5%.

Sarcopenia was associated with older age, men, low BMI, poor functional status, hospital stays in previous months, and certain chronic conditions like chronic lung disease, neurological diseases, and neoplasia. All of these factors have been involved and associated with sarcopenia in previous studies, especially a low BMI, poor functional status, and severe chronic diseases [37,38].

Frailty was present in more than 60% of our patients, and increased to more than 90% when pre-frail patients were also included. This prevalence was higher than those detected recently by other authors in different elderly populations [35,36,40,41]. This difference may be explained by the high disease burden of our cohort of PP and its hospital-based nature, whereas most previous studies were centered in community-dwelling or nursing home populations.

An interesting issue emerging from our data is the notably higher prevalence of frailty with respect to sarcopenia. Undoubtedly, these two syndromes are deeply related, and share phenotypes. As a matter of fact, 18% of our patients suffered both; that is, that most sarcopenic patients were frail, but only around one-third of frail patients were sarcopenic; this behavior has also been detected in other studies [42]. Nevertheless, these data contrast with the classic rationale of aging and diseases, which promote muscle wasting, sarcopenia development, and finally clinical frailty appearance [43]. Frequently, frailty is more age related, whereas sarcopenia is also related to disease, starvation, and disuse, but patients in the present study were not extremely old and suffered from severe chronic diseases, so this explanation does not fit well with our observations. On the other hand, criteria defining the two conditions overlap, but frailty requires weight loss, whereas sarcopenia requires muscle loss. We think that the more strict view in muscle wasting of sarcopenia definition may underestimate patients in early muscle-wasting stages, whereas frailty definition could be more sensitive in detecting these patients at the risk of weakness and muscle-wasting [44,45].

Frailty was associated with women, poor functional status, more severe polypathology, chronic neurological conditions, pain, asthenia and anxiety disorders. These factors are similar to those found previously by other authors [9,10,11,46]. Strikingly frailty was associated with women, while sarcopenia to men; this finding has already been reported, and its possible explanations are diverse (social factors, life and physical exercise habits, and subjective differences of "exhaustion" subjective feeling between men and women) [9,10,11,46].

The 12-month mortality rate was high, but similar to that observed in previous hospital-based cohorts [47]. Cumulative survival was significantly lower in patients with sarcopenia, frailty, or both with respect to those without these conditions. These data are in concordance with multiple previous observations, in which sarcopenia and frailty were strong predictors of morbidity and mortality in other populations [27,28,29,30,36,37,38,39]. It may be difficult to outline the causative role of polypathology/multimorbidity in the development of sarcopenia/frailty in vulnerable populations or the opposite, probably both play mutual influences in their development and outcomes. In this sense, focusing attention on these two prevalent conditions in polypathological patients, and establishing specific interventions, will probably benefit and improve the health-related outcomes of this emergent population.

Finally, this study has some limitations. Most patients were recruited in-hospital, so it is possible that it may not be applicable to other clinical scenarios like hospices, or primary care settings. Another possible limitation is the use of BIA for assessing skeletal muscle mass; it is known that BIA could lose accuracy in the presence of different conditions like edema, and is not useful in amputated patients; for this reason, we performed this measurement to all patients in the best clinical stability conditions.

## 5. Conclusions

In conclusion, sarcopenia and frailty are frequent and interrelated conditions in polypathological patients, shadowing their survival. Their early recognition and management could improve health-related outcomes in this population.

## Figures and Tables

**Figure 1 jcm-08-00535-f001:**
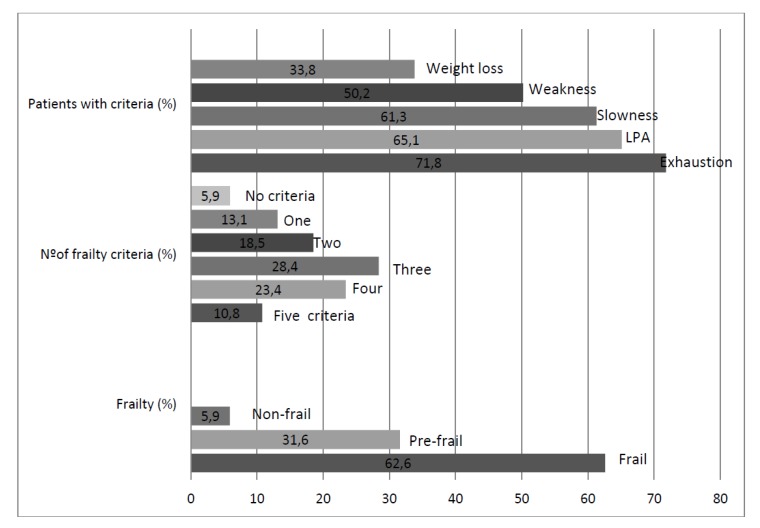
Prevalence of different aspects of frailty in a multicenter prospective cohort of polypathological patients. LPA: low physical activity.

**Figure 2 jcm-08-00535-f002:**
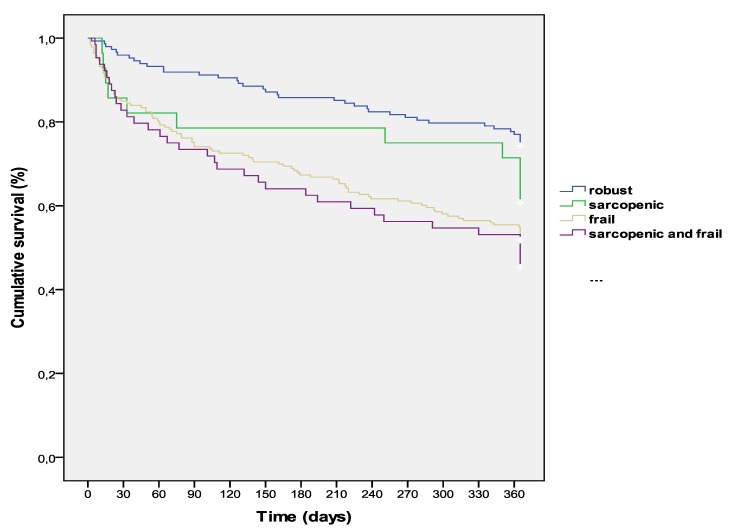
Kaplan–Meier 12-month survival curves of a multicenter cohort of polypathological patients according to the presence of sarcopenia, frailty, or both.

**Table 1 jcm-08-00535-t001:** Functional definition of Polypathological Patient: the patient who suffers chronic diseases included in two or more of the following clinical categories.

**Category A**
A.1 Chronic heart failure with past/present stage II dyspnea of NYHA ^1^ A.2 Coronary heart disease
**Category B**
B.1 Vasculitides and/or systemic autoimmune diseases B.2 Chronic renal disease (creatininaemia >1.4/1.3 mg/dL in men/women or proteinuria ^2^, during ≥3 months
**Category C**
Chronic lung disease with past/present stage 2 dyspnea of MRC ^3^, or FEV1 < 65%, or basal SatO_2_ ≤ 90%
**Category D**
D.1 Chronic inflammatory bowel disease D.2 Chronic liver disease with evidence of portal hypertension ^4^
**Category E**
E.1 Stroke E.2 Neurological disease with permanent motor deficit, leading to severe impairment of basic activities of daily living (Barthel’s index < 60). E.3 Neurological disease with permanent moderate-severe cognitive impairment (Pfeiffer’s test with ≥5 errors).
**Category F**
F.1 Symptomatic peripheral artery disease F.2 Diabetes mellitus with proliferate retinopathy or symptomatic neuropathy
**Category G**
G.1 Chronic anemia (Hb < 10g/dL during ≥3 months) due to digestive-tract losses or acquired haemopathy not tributary of treatment with curative intention G.2 Solid-organ or Hematological active neoplasia not tributary of treatment with curative intention
**Category H**
Chronic osteoarticular disease, leading to severe impairment of basic activities of daily living (Barthel’s index < 60)

^1^ Slight limitation of physical activity. Comfortable at rest, but ordinary physical activity results in fatigue, palpitation, or dyspnea. ^2^ Albumin/Creatinine index > 300 mg/g, microalbuminuria > 3 mg/dL in urine, albumin > 300 mg/day in 24-h urine, or albuminuria/min > 200 microg/min. ^3^ Short of breath when hurrying or walking up a slight hill. ^4^ Presence of clinical, analytical, sonographic, or endoscopic data of portal hypertension.

**Table 2 jcm-08-00535-t002:** Main clinical features of a multicenter sample of 444 polypathological patients recruited for sarcopenia and frailty assessment.

Clinical Features	Mean (SD)/Median [IQR]/*N* (%)
Number of defining categories per patient	2.5 (0.8)
Patients with ≥ 3 categories	175 (39.5%)
Prevalence of defining categories in recruited polypathological patients	
Category A (heart diseases)	374 (84.6%)
Category B (kidney/autoimmune diseases)	202 (45.7%)
Category C (lung diseases)	183 (41.4%)
Category E (neurological diseases)	133 (30.1%)
Category F (peripheral arterial disease/diabetes with neuropathy)	80 (18.1%)
Category G (chronic neoplasia/anemia)	70 (15.8%),
Category H (degenerative osteoarticular disease)	43 (9.7%)
Category D (liver disease)	28 (6.3%)
Number of other comorbidities per patient	5.9 (2.3)
Cardiovascular	1.8 (0.9)
Endocrine and metabolic	1.6 (1)
Respiratory	0.75 (0.9)
Most frequent comorbidities	
Hypertension	380 (86%)
Dyslipemia	232 (52.5%)
Diabetes with no visceral involvement	216 (49%)
Atrial fibrillation	178 (40%)
Obesity	159 (36%)
Anxiety and depressive disorders	74 (17%)
Benign prostate hyperplasia	64 (14.5%)
Osteoporosis	42 (9.5%)
Frequent symptoms	
Fatigue	304 (70%)
Anorexia	212 (48%)
Insomnia	194 (44%)
Chronic pain	178 (40%)
Cough	158 (36%)
Patients with basal III-IV class of NYHA // III-IV class of mMRC	128 (29%)
Patients with delirium in last hospital admission	76 (17%)
Nausea/Vomiting	37 (8.5%)
Pressure ulcer(s)	35 (8%)
PROFUND index	6 [6]
Number of prescribed drugs at inclusion / Patients with polypharmacy	10 (4)/429 (96.5%)
Patients with home oxygen therapy	74 (17%)
Hospitalizations in last 3 months / total days in hospital in last 3 months	0.6 (0.8)/5 (9)
Basal Barthel´s Index	66 (30)

SD: standard deviation; IQR: interquartile range; NYHA: New York Heart Association; mMRC: Medical Research Council.

**Table 3 jcm-08-00535-t003:** Main biological and anthropometric parameters of a multicenter sample of 444 polypathological patients recruited for sarcopenia and frailty assessment.

Biological and Anthropometric Features	Mean (SD)/Median [IQR]
Main biological parameters	
Hemoglobin (d/dL)	11.3 (2)
Creatinin (mg/dL)	1.26 (1)
Albumin (g/dL)	3.2 (0.9)
Bilirrubin (mg/dL)	0.47 [0.6]
Sodium (mEq/L)	139 (8)
Calcium (mg/dL)	8.7 (0.7)
Cholesterol (mg/dL)	151 (42)
Triglicerydes (mg/dL)	116 (80)
Ferritin (ng/mL)	105 (211)
Vitamin D (ng/mL)	11 (17)
Leucocytes (number/µL)	8000 (4000)
Lymphocytes (number/µL)	1200 (400)
Anthropometric features BMI (kg/m^2^)	30 (6.6%)
Dominant hand strength (kg)	18 (16)
Men	27 (16)
Women	14 (10)
Patients with dominant hand strength below 50 percentile	223 (50.5%)
Men	112 (46%)
Women	111 (56%)
Skeletal muscle mass index (kg/m^2^)	11.9 (4.8)
Men	12.9 (5)
Women	10,9 (4)
Total body water (L)	42 (10)
Men	46 (109
Women	37 (8)
Total Fat mass (kg)	26 (13)
Men	23.1 (12)
Women	29 (14)

SD: standard deviation; IQR: interquartile range; BMI: body mass index.

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
