# Peer review of "Impact of Sarcopenia and Frailty in a Multicenter Cohort of Polypathological Patients"

_jcm, 2019, doi:10.3390/jcm8040535_

Reviewer 1 Report

This study shows that sarcopenia and frailty are often present with poly-pathologies and associated with reduced survival. It is not clear what is cause and effect, though I suspect the sarcopenia/frailty is the effect of the pathology. So, frailty/sarcopenia may help as a detector of these conditions, but rather than, as the title says, the impact of sarcopenia, it is the impact of those other things that cause sarcopenia/frailty and a poor outcome. In fact, probably increasing muscle mass would not help these people much. Could you rephrase along the lines that it may be an indicator/early hallmark of associated problems?

Page 4 line 129-130 is a repeat of the previous sentence.

Page 4 line 146 ‘Worl’

Page 8 line 181 ‘low’ rather than ‘slow’ handgrip strength

Page 11 line 245-247 I have no idea what you want to tell the reader here.

Page 11 line 249 ‘A more sensitive-less’ ??? What is that?

Page 11 first paragraph I find hard to follow. Can you be more specific in telling what is good in EWGSOP and FNIH and why EWGSOP is the better of the two in your opinion? Also, consider why it matters for your study. You do so, but it is all rather vague to me.

Author Response

Thank you for your kind comments about our work.

We agree in part with the general comment, polypathology and multimorbitidy probably enhance and worsen frailty and sarcopenia; but for sure age-related frailty and sarcopenia overshadow survival and health-related results of polypathological patients, as shown in our data. Nevertheless,  according to reviewer`s suggestion we have rephrased along the lines that it may be an indicator/early hallmark of associated problems.

Lines 291-293: ). It may be difficult to delimit the causative role of polypathology/multimorbidity in the development of sarcopenia/frailty in vulnerable populations or the opposite; probably both play mutual influences in their development and outcomes. I

We have amended all details and suggestions regarding Pages 4, 8, and 11

With respect to the differences among different criteria sets we have also incorporated our opinion referring the advantages and disadvantages of them.

Yours sincerely. MBW

Reviewer 2 Report

Even if multiple cohorts on sarcopenia already exists, this paper is original because focus on a population of polypathological patients.

The manuscript is well written, well organized, very clear and the statistics have been correctly performed.

I only have small concerns about this manuscript:

-          In methods, the part ‘reference population’ is unclear and deserve additional explanations;

-          Authors indicated that they assessed functional parameters but only reported Barthel’s index. Are there other tests performed?

-          Why did the authors not follow the standardized protocol published by Roberts et al. for the assessment of muscle strength?

-          A revised version of the EWGSOP criteria has recently been published. Authors should discuss it in the discussion. If possible, it could be interesting to look at the results when using this new definition of sarcopenia instead of the previous one. The manuscript would be more original.

-          Please be more specific about the ‘calculation’ of muscle mass with BIA.

-          Were the BIA assessments performed with fasted subjects? Was the BIA calibrated prior assessments?

-          How was gait speed measured? With which protocol?

-          Authors wrote that the data were collected by different clinicians. How many clinicians exactly? How could they ensure a standardization of the measurements with different clinicians?

-          How did the authors deal with missing data?

-          Table 3. There is a problem in the presentation of results of men and women. Only one result appears, and it is interesting to have both results of men and women. Please also provide results separately for men and women for fat mass.

Author Response

Thank you for your kind comments about our work.

- Following reviewer's suggestion we have incorporated more details to 'reference population'.

- Functional assesssment: With respect to Activities of Daily Living the most widespread and universal scale is Barthel's index; for this reason we selected this scale for the assessment of ADL. We did not assess instrumental activities because many of our patients already presented some difficulties in performing ADL.

- We fully agree with this reviewer's comment. As a matter of fact we followed Roberts et al. protocol, but with a slight modification (two attempts per hand instead of three, and selection of 'best attempt'. We have included this comment in the revised version, and we have also included the reference.

- Gait speed assessment was performed following EWGSOP recommendations, specifically those used by Laurentani et al. We only modified this protocol substituting the photocells with a 4m colored adhesive ribbon, which was adhered to the ground. Participants were instructed to stand with both feet touching the ribbon start and to begin walking at their usual pace after a verbal command, until the end of the ribbon. The time between the start and end was recorded by investigator with a digital chronometer. The average of two walks was used to compute a measure of walking speed. According to reviewer's suggestion we have incorporated this detailed description in the paper, and the reference.

- Data were collected by a total of 8 clinicians. All clinicians were experts in assessing and treating elderly patients, and had worked previously in assessing muscle strength, muscle mass and frailty. In the preparation of the study all investigators were specifically instructed with the aim to homogenously incorporate patients' data.

- There were very scarce missing data in the study. Most missing data were secondary variables, and none of the main variables were missing.

- According to reviewer's suggestion we have also incorporated results separately for men and women for fat mass.

Yours sincerely. MBW

Reviewer 3 Report

Thank you for the opportunity to review this work which examines the prevalence of sarcopenia and frailty in a cohort of polypathological patients. Moreover, the study also examines the independent association of various factors associated with these geriatric conditions.

Despite the results being interesting and of clinical significance, regrettably the write up and clarity of the manuscript, particularly the methodology, was not to standard and requires major revision. For example, lines 43-45 in the introduction are simply unclear? Moreover, lines 58 and 67 (prevalence of frailty) are without reference.

The methodology (protocol) around BIA, muscle strength and physical performance lacks the appropriate level of detail. The methodology also gives reference to identification of cachexia - why? This was unrelated to the study, nor used in the analysis. If it was, this was unclear. Moreover, using BMI (<20) in isolation is not an appropriate diagnostic criteria for identification of cachexia. The authors should carefully check the diagnostic criterion cut-offs for identification of sarcopenia (in particular low muscle strength) as these appear incorrect in the manuscript? Lastly, given the recent revision of the diagnostic criteria for identification of sarcopenia in 2018 (EWGSOP2), the authors should consider using the revised guidelines and criteria in their statistical analysis.    

Author Response

Thank you for your kind comments about our work.

- Following reviewer's suggestion we have incorporated more details in introduction section to clarify this issue to readers, and we have also incorporated additional references to frailty prevalence.

- According to these reviewer's suggestion we have included more specific aspects of physical performance and muscle strength. We have also deleted data and comments regarding cachexia in order to fulfill reviewer's recommendation.

- With respect to muscle strength assessment wue have specified the cut-off pints following  EWGSOP recommendations.

- Finally we fully agree with this reviewer in his-her comment about EWGSOP2 criteria, which have been published in January 2019. Unfortunately, this study was designed and performed from 2012 to 2016, and followed, for this reason EWGSOP criteria. Nevertheless, and according to reviewer's comment we have incorporated a paragraph in the Discussion section referring to EWGSOP2 criteria.

Sincerely yours. MBW